# Tgfb3 collaborates with PP2A and notch signaling pathways to inhibit retina regeneration

**Mi-Sun Lee†, Jin Wan†, Daniel Goldman\***

Michigan Neuroscience Institute and Department of Biological Chemistry, University of Michigan, Ann Arbor, United States

**Abstract** Neuronal degeneration in the zebrafish retina stimulates Müller glia (MG) to proliferate and generate multipotent progenitors for retinal repair. Controlling this proliferation is critical to successful regeneration. Previous studies reported that retinal injury stimulates pSmad3 signaling in injury-responsive MG. Contrary to these findings, we report pSmad3 expression is restricted to quiescent MG and suppressed in injury-responsive MG. Our data indicates that Tgfb3 is the ligand responsible for regulating pSmad3 expression. Remarkably, although overexpression of either Tgfb1b or Tgfb3 can stimulate pSmad3 expression in the injured retina, only Tgfb3 inhibits injury-dependent MG proliferation; suggesting the involvement of a non-canonical Tgfb signaling pathway. Furthermore, inhibition of Alk5, PP2A or Notch signaling rescues MG proliferation in Tgfb3 overexpressing zebrafish. Finally, we report that this Tgfb3 signaling pathway is active in zebrafish MG, but not those in mice, which may contribute to the different regenerative capabilities of MG from fish and mammals.

**\*For correspondence:**
neuroman@umich.edu

†These authors contributed equally to this work

**Competing interests:** The authors declare that no competing interests exist.

## Introduction

Blinding eye diseases like macular degeneration, glaucoma, and diabetic retinopathy result in retinal neuron death which leads to vision loss. Unlike mammals, zebrafish have a remarkable ability to regenerate neurons that were lost due to injury or disease (*Goldman, 2014*; *Lenkowski and Raymond, 2014*; *Wan and Goldman, 2016*). Key to this regenerative response are Müller glia (MG) the major glial cell type in the retina of both fish and mammals. The normal function of MG is to maintain retinal structure and homeostasis (*Bringmann et al., 2009*; *MacDonald et al., 2015*; *Reichenbach and Bringmann, 2013*). However, in fish, MG respond to retinal injury by dividing and generating multipotent progenitors for neuron regeneration (*Bernardos et al., 2007*; *Fausett and Goldman, 2006*; *Fimbel et al., 2007*; *Powell et al., 2016*; *Ramachandran et al., 2010b*; *Raymond et al., 2006*).

Although it is not known why MG from fish and mammals respond differently to retinal injury, it likely results from differences in their environment and intrinsic differences that are reflected in their gene expression programs. Unlike mammals, the zebrafish retina continues to grow throughout life and this growth-permissive environment may impact MG's potential to mount a regenerative response (*Hitchcock and Raymond, 2004*). Zebrafish MG themselves also appear to contribute to a pro-regenerative environment by releasing growth factors and cytokines after retinal injury (*Calinescu et al., 2009*; *Kaur et al., 2018*; *Nelson et al., 2013*; *Ramachandran et al., 2011*; *Wan et al., 2012*; *Wan et al., 2014*; *Zhao et al., 2014*). Dying neurons and immune cells may also contribute to the pro-regenerative MG niche in fish. In addition to niche factors, intrinsic differences in gene expression programs in MG from fish and mammals have been noted (*Sifuentes et al., 2016*). In fish, Ascl1a and Lin28a are critical factors promoting MG reprogramming and proliferation (*Elsaeidi et al., 2018*; *Fausett et al., 2008*; *Gorsuch et al., 2017*; *Mitra et al., 2018*; *Mitra et al.,*

*2019*; *Ramachandran et al., 2010a*; *Ramachandran et al., 2011*). *ascl1a* and *lin28a* RNAs are highly induced in MG following injury to the fish retina, while their homologs remain undetectable in the injured mouse retina (*Elsaeidi et al., 2018*; *Karl et al., 2008*). However, forced expression of Ascl1, along with HDAC inhibition or Lin28a expression can stimulate a limited proliferative response by MG in the injured mouse retina (*Elsaeidi et al., 2018*; *Jorstad et al., 2017*).

Another intrinsic difference between MG from fish and mammals is Notch signaling. In mice, Notch signaling declines as MG differentiate and mature; while in fish, Notch signaling is maintained into adulthood (*Bernardos et al., 2005*; *Dorsky et al., 1995*; *Elsaeidi et al., 2018*; *Furukawa et al., 2000*; *Nelson et al., 2011*; *Wan and Goldman, 2017*; *Wan et al., 2012*). The maintenance of Notch signaling in MG of the adult zebrafish retina contributes to MG quiescence and Notch suppression is required for MG proliferation (*Conner et al., 2014*; *Elsaeidi et al., 2018*; *Taylor et al., 2015*; *Wan and Goldman, 2017*; *Wan et al., 2012*). Furthermore, the opposing actions of Fgf8a on MG proliferation in juvenile and adult fish is correlated with corresponding changes in Notch signaling activity (*Wan and Goldman, 2017*). Thus, Notch signaling is a major control point in the decision to proliferate or remain quiescent and understanding how Notch signaling is regulated in the zebrafish retina will help reveal mechanisms underlying MG's decision to mount a regenerative response.

In addition to Notch signaling, Tgfb signaling has been implicated in regulating injury-dependent MG proliferation in the zebrafish retina (*Conedera et al., 2020*; *Lenkowski et al., 2013*; *Sharma et al., 2019*; *Sharma et al., 2020*; *Tappeiner et al., 2016*). However, there are inconsistencies among these reports with some suggesting it is inhibited in proliferating MG (*Lenkowski et al., 2013*; *Sharma et al., 2019*) and others suggesting it is activated in these cells (*Conedera et al., 2020*; *Sharma et al., 2020*; *Tappeiner et al., 2016*). Although most of the above studies suggest Tgfb signaling inhibits MG proliferation, one study suggests it is necessary for injury-dependent MG proliferation (*Sharma et al., 2020*). Besides these inconsistencies, the endogenous ligand responsible for stimulating Tgfb signaling and the downstream signaling components responsible for regulating MG proliferation remain unknown.

In zebrafish, Tgfb ligands are encoded by four genes: *tgfb1a*, *tgfb1b*, *tgfb2*, and *tgfb3*. Canonical Tgfb signaling occurs when a Tgfb ligand engages a type II receptor that recruits type I (Alk5) receptor to stimulate phosphorylation of Smad2 and Smad3, which stimulates their nuclear import and allows for regulation of target genes (*Shi and Massagué, 2003*). Non-canonical Tgfb signaling refers to receptors that engage Erk, Jnk, p38, or protein phosphatase 2A (PP2A)-dependent pathways (*Derynck and Zhang, 2003*; *Petritsch et al., 2000*). Identification of the specific Tgfb ligands mediating retina regeneration in fish and unravelling their mechanism of action are critical for understanding how the Tgfb signaling pathway regulates MG proliferation and retina regeneration.

Here we provide evidence indicating Tgfb3 controls MG quiescence via a non-canonical Tgfb signaling pathway. Of all Tgfb ligand encoding genes, we find *tgfb3* expression is uniquely restricted to quiescent MG in the adult zebrafish retina. Following retinal injury, this expression is suppressed at the injury site. Using transgenic fish that allow for conditional expression of Tgfb3, we show that *tgfb3* suppression is necessary for injury-dependent MG proliferation. Interestingly, our studies reveal a specificity in the actions of Tgfb ligands on MG proliferation with Tgfb3, but not Tgfb1b, stimulating MG quiescence. Our studies suggest PP2A and Notch signaling pathways act downstream of Tgfb3. Furthermore, we report that Tgfb3 stimulates pSmad3 expression in the injured retina; however, pSmad3 expression is not sufficient to drive MG quiescence. Finally, we report that the Tgfb3 expression is not detectable in mouse MG, and this may contribute to their poor regenerative potential.

## Results

### pSmad3 signaling is suppressed in injury-responsive MG

pSmad3 immunofluorescence was used to detect canonical Tgfb signaling in the uninjured retina of *gfap:GFP* transgenic fish. This analysis revealed that pSmad3 expression was restricted to GFP+ MG (*Figure 1A*). Importantly, this expression was suppressed when fish were immersed in water containing the Tgfb receptor 1 (Alk5) kinase inhibitors SB431542 or SB505124 (*Figure 1—figure supplement 1A*; *Vogt et al., 2011*).

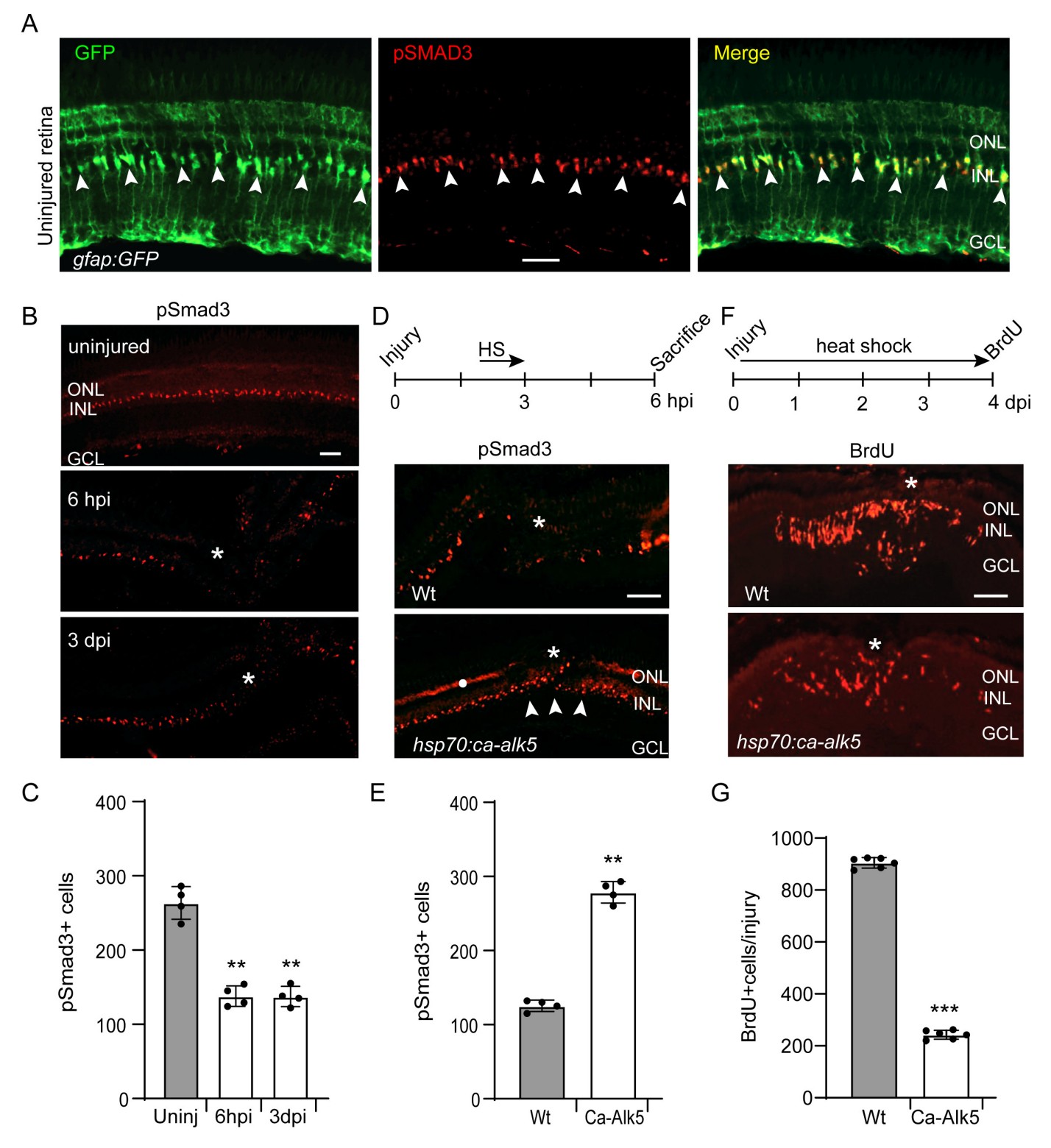

**Figure 1.** pSmad3 expression in the uninjured and injured retina. (**A**) Retinal section from uninjured *gfap:GFP* fish retina with GFP (green) and pSmad3 (red) immunofluorescence. Arrowheads point to pSmad3 expressing MG. (**B**) pSmad3 immunofluorescence in uninjured and needle poke injured retina. Asterisk marks injury site. (**C**) Quantification of data shown in (**B**). (**D**) Top diagram shows time line for heat shock treatment after injury and when fish were sacrificed. Bottom panels show pSmad3 immunofluorescence in injured and heat shock-treated Wt and *hsp70:ca-Alk5* transgenic fish. Asterisk marks the injury site and arrowheads point to recovery of pSmad3 expression at the injury site in heat shock-treated *hsp70:ca-alk5* transgenic fish. White

*Figure 1 continued on next page*

*Figure 1 continued*

dot in lower panel marks non-specific autofluorescence in the photoreceptor layer. (E) Quantification of data shown in (D). (F) Top diagram is time line for experiment illustrating injury, heat shock treatment and BrdU labelling prior to sacrifice. Lower panels show BrdU immunofluorescence in Wt and *hsp70:ca-alk5* fish. Asterisk marks the injury site. (G) Quantification of data presented in (F). Error bars are SD. **p<0.01, ***p<0.001. Scale bar is 50 microns.

The online version of this article includes the following figure supplement(s) for figure 1:

**Figure supplement 1.** Alk5-dependent pSmad3 expression.

We next investigated if pSmad3 expression was regulated by retinal injury. A needle poke was used to cause a focal injury (*Fausett and Goldman, 2006*). This manipulation stimulated a rapid depletion in pSmad3 expression at the injury site (*Figure 1B–C*), which was rescued after heat shock of *hsp70:ca-Alk5* transgenic fish that express a constitutively active Tgfb receptor 1 (ca-Alk5, T204D) under the control of a *hsp70* heat shock promoter (*Figure 1D–E*; *Wieser et al., 1995*; *Zhou et al., 2011*). Furthermore, forced expression of ca-Alk5 inhibited injury-dependent MG proliferation (*Figure 1F–G*). However, when TUNEL stain was used to identify apoptotic cells, very few TUNEL+ cells were identified (*Figure 1—figure supplement 1B*). Thus, the reduced injury-dependent MG proliferation noted with ca-Alk5 overexpression may reflect increased pSmad signaling and/or reduced cell death. Regardless, these data indicate that Tgfb signaling via pSmad3 expression correlates with MG proliferation in the injured retina.

## *tgfb3* expression correlates with injury-dependent pSmad3 expression

To identify injury-responsive Tgfb ligands that might regulate pSmad3 signaling in the retina, we interrogated RNAseq data sets from MG and MG-derived progenitors that were isolated from uninjured *gfap:GFP* and injured *1016 tuba1a:GFP* transgenic fish retinas, respectively (*Fausett and Goldman, 2006*; *Kassen et al., 2007*). This analysis indicated constitutive and low expression of *tgfb1a*; injury-dependent induction of *tgfb1b* and *tgfb2*; and injury-dependent suppression of *tgfb3* (*Figure 2A*). qPCR was used to validate the RNAseq data (*Figure 2B–C*; *Figure 2—figure supplement 1A*). Injury-dependent reduction in *tgfb3* RNA was also evident following NMDA-mediated amacrine and ganglion cell death or genetic ablation of photoreceptors using metronidazole-treated *zop:nsfb-EGFP* transgenic fish that harbor a zebrafish rod opsin promoter driving expression of bacterial nitroreductase that converts metronidazole into a cytotoxic product (*Figure 2—figure supplement 1B*; *Montgomery et al., 2010*; *Powell et al., 2016*).

We also compared the relative expression of *tgfb* ligand encoding RNAs in the retinal neuron (GFP-) and MG (GFP+) cell populations that were separated by FACS using dissociated cells from uninjured *gfap:GFP* fish retinas. This analysis showed *tgfb3* is highly enriched in MG, whereas *tgfb1b* and *tgfb2* are more equally distributed between these different cell populations (*Figure 2—figure supplement 1C*). *In situ* hybridization assays for *tgfb3* combined with glutamine synthetase (GS) immunofluorescence on retinal sections confirmed *tgfb3* gene expression is restricted to GS+ MG in the uninjured retina (*Figure 2D*).

We next used *in situ* hybridization assays to investigate when during development this MG-specific *tgfb3* expression profile emerges. Consistent with a previous report (*Cheah et al., 2005*), we found transient expression of *tgfb3* RNA in the lens at 24 hpf (hours post fertilization) (*Figure 3A*). At 96 hpf when MG are differentiating (*Bernardos et al., 2005*), we do not detect significant levels of *tgfb3* RNA in the retina, nor do we observe significant *tgfb3* expression at 7 dpf (days post fertilization). By 10 dpf, *tgfb3* RNA is detected in the central region of the retina where more mature MG reside (*Figure 3A*). This expression continues to increase with age and the adult expression pattern is established by 3 mpf (months post fertilization) (*Figure 3A*). Consistent with our RNAseq and qPCR data indicating very low *tgfb1b* levels in the retina (*Figure 1A–B*), we were unable to detect zebrafish *tgfb1b* gene expression by *in situ* hybridization at any of the time points examined (*Figure 3A*).

We next examined *Tgfb3* expression in the 1 month old mouse retina. This analysis showed *Tgfb3* expression was confined to cells in the GCL and INL (*Figure 3B*). Co-staining retinal sections for *Tgfb3* RNA and Sox9 immunofluorescence (MG marker) showed no overlap, indicating *Tgfb3+* cells in the INL are not MG (*Figure 3B*).

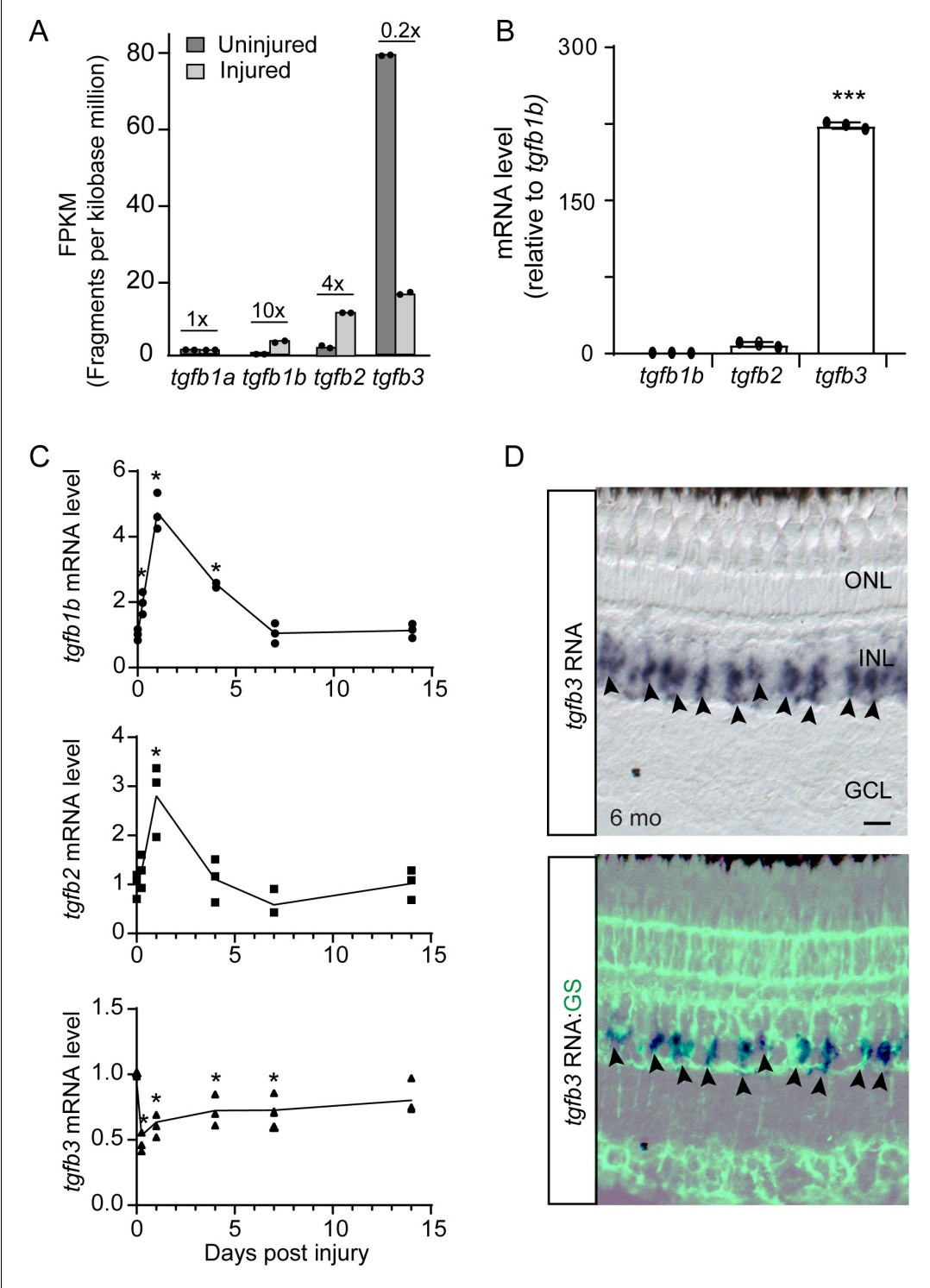

**Figure 2.** Injury-dependent regulation of *tgfb* gene expression. (**A**) RNAseq was used to quantify *tgfb* gene expression in FACS purified GFP+ MG isolated from uninjured and injured (2 dpi) *gfap:GFP* and *1016 tuba1a:GFP* fish retinas, respectively. Fold change in gene expression following retinal injury is indicated above the bars. (**B**) qPCR quantification of *tgfb* gene expression in GFP+ MG FACS purified from uninjured *gfap:GFP* fish retina (normalized to *tgfb1b*). (**C**) qPCR quantification of *tgfb* gene expression following retinal injury normalized to uninjured levels. Total retinal RNA was used for qPCR. Error bars are SD. *p<0.05. (**D**) *tgfb3 in situ* hybridization and GS (glutamine synthetase) immunofluorescence in adult fish retina shows *tgfb3* RNA is expressed in MG. Top panel is *tgfb3 in situ* hybridization; bottom panel is overlay of *in situ* hybridization and GS immunofluorescence. Arrowheads point to *tgfb3*-expressing MG. Scale bar is 50 microns. *p<0.05, ***p<0.001.

*Figure 2 continued on next page*

*Figure 2 continued*

The online version of this article includes the following figure supplement(s) for figure 2:

**Figure supplement 1.** *tgfb* gene expression in uninjured and injured retina.

Although the lack of detectable Tgfb3 expression in the developing (1–7 dpf) fish retina suggests Tgfb3 expression is not necessary for MG differentiation, low levels of expression may go undetected by *in situ* hybridization. Therefore, we investigated if knocking down Tgfb3 expression with a translation-blocking morpholino-modified antisense oligonucleotide (MO) would affect MG differentiation. Control or *tgfb3*-MOs were delivered into single cell zebrafish embryos and at 6 dpf, fish were sacrificed and retinal sections assayed for glutamine synthetase (GS) immunofluorescence which serves as a marker of differentiated MG. This analysis revealed normal GS reactivity in the developing retina regardless of Tgfb3 knockdown (*Figure 3—figure supplement 1A*). We confirmed the *tgfb3*-MO's effectiveness by observing reduced GFP in embryos injected with a *tgfb3-EGFP* fusion RNA and the *tgfb3*-MO, but not with control MO (*Figure 3—figure supplement 1B*).

Because of concerns that MO-mediated Tgfb3 knockdown may not be effective at 6 dpf due to MO dilution during cell division, we generated *tgfb3⁻/⁻* fish using a CRISPR/Cas9 strategy (*Figure 3—figure supplement 1C–E*). In these fish, an insertion mutation changes the *tgfb3* reading frame so a premature stop codon is introduced in exon 1 (*Figure 3—figure supplement 1E*). We found that *tgfb3⁻/⁻* fish die around 2 weeks post fertilization. Therefore, we selected *tgfb3⁺/⁺*, *tgfb3⁺/⁻* and *tgfb3⁻/⁻* fish at 12 dpf to assay for retinal GS expression. Normal GS immunofluorescence was observed in both *tgfb3⁺/⁻* and *tgfb3⁻/⁻* fish retinas suggesting Tgfb3 does not impact MG differentiation (*Figure 3—figure supplement 1F*). However, we do note that retinas were smaller in *tgfb3⁻/⁻* fish.

We next used *in situ* hybridization assays to investigate the spatial pattern of *tgfb3* expression in the injured adult retina. This analysis revealed that *tgfb3* RNA is specifically suppressed at the injury site and that this suppression preceded MG proliferation that begins ~2 dpi (*Figure 3C*).

Together, the above results indicate that *tgfb3* expression in the zebrafish retina correlates with MG maturation, pSmad3 expression, and MG quiescence.

## Injury-dependent Tgfb3 suppression is required for MG proliferation

To investigate if Tgfb3 suppression was required for injury-dependent MG proliferation, we generated *hsp70:tgfb3* transgenic fish that allow conditional expression of Tgfb3 with heat shock. A 1 hr heat shock at 37°C resulted in over a 100-fold induction of *tgfb3* RNA that persisted for over 3 hr and then returned to basal levels around 24 hr post heat shock (*Figure 4—figure supplement 1A*). To examine the effect Tgfb3 had on MG proliferation in the injured retina, *hsp70:tgfb3* transgenic fish received a needle poke injury and then a 1 hr heat shock every 6 hr for 4 days. Three hours before sacrifice at 4 dpi, fish received an IP (intraperitoneal) injection of BrdU to label proliferating cells. This analysis showed that forced expression of Tgfb3 suppressed MG proliferation in the injured retina (*Figure 4A–B*), without any significant effect on injury-dependent cell death (*Figure 6—figure supplement 1B*).

We next investigated if Tgfb3 depletion affected spontaneous or injury-dependent MG proliferation. Because *tgfb3⁻/⁻* fish do not survive to adults, we knocked down Tgfb3 with a *tgfb3*-targeting MO and determined if there was an effect on MG proliferation. Control or *tgfb3*-MO was delivered intravitreally and cellular uptake facilitated by electroporation as previously described (*Thummel et al., 2011*). Retinas were injured and at 3 dpi fish received an IP injection of EdU 3 hr prior to sacrifice. Quantification of EdU+ and TUNEL+ cells revealed Tgfb3 knockdown had no effect on MG proliferation or cell death at either the injury site or in undamaged regions of the retina (*Figure 4—figure supplement 1B–C*).

Together, the above data indicate Tgfb3 suppression is necessary for injury-dependent MG proliferation, but this suppression is not sufficient to drive MG proliferation in the uninjured retina.

## Tgfb3 regulates MG reprogramming

Reprogramming MG for retinal repair requires the regulation of gene expression programs that stimulate MG proliferation and the expansion of a MG-derived progenitor population. Essential

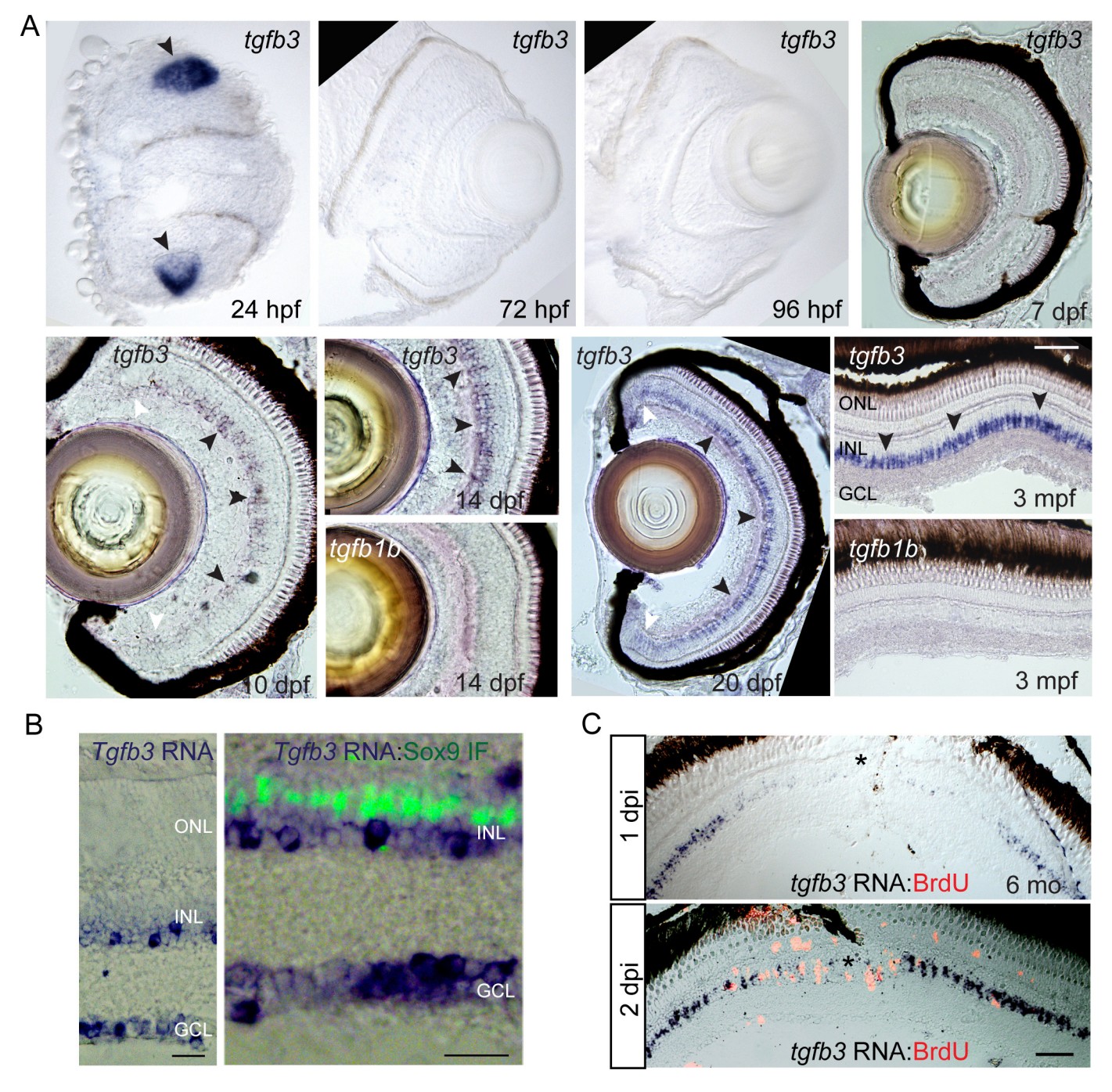

**Figure 3.** *tgfb3* expression in developing and adult retina. (**A**) *tgfb3 in situ* hybridization identifies *tgfb3* expression in lens at 24 hpf (hours post fertilization) and in MG beginning ~10 dpf (days post fertilization). This latter expression continues to increase throughout the first 3 months of development. Note *tgfb1b* expression remains undetectable by *in situ* hybridization at all the time points examined (14 dpf and 3 mpf). Size marker is 20 microns. In the 10 and 20 dpf panels, black arrowheads point to *tgfb3* expressing MG in the central retina, while white arrowheads point to reduced *tgfb3* expression in the retinal periphery. (**B**) *In situ* hybridization (blue/purple product) and Sox9 immunofluorescence (green fluorescence) identifies *tgfb3* expression in GCL and INL, but not in MG of the mouse retina. Size marker is 25 microns. (**C**) Retinal injury suppresses *tgfb3* expression at the injury site in 3 month old fish. *In situ* hybridization detects *tgfb3* RNA (blue/purple product) and BrdU immunofluorescence (red/orange fluorescence) identifies proliferating cells. Size marker is 40 microns.

The online version of this article includes the following figure supplement(s) for figure 3:

**Figure supplement 1.** Tgfb3 knockdown and *tgfb3* gene editing do not affect MG differentiation.

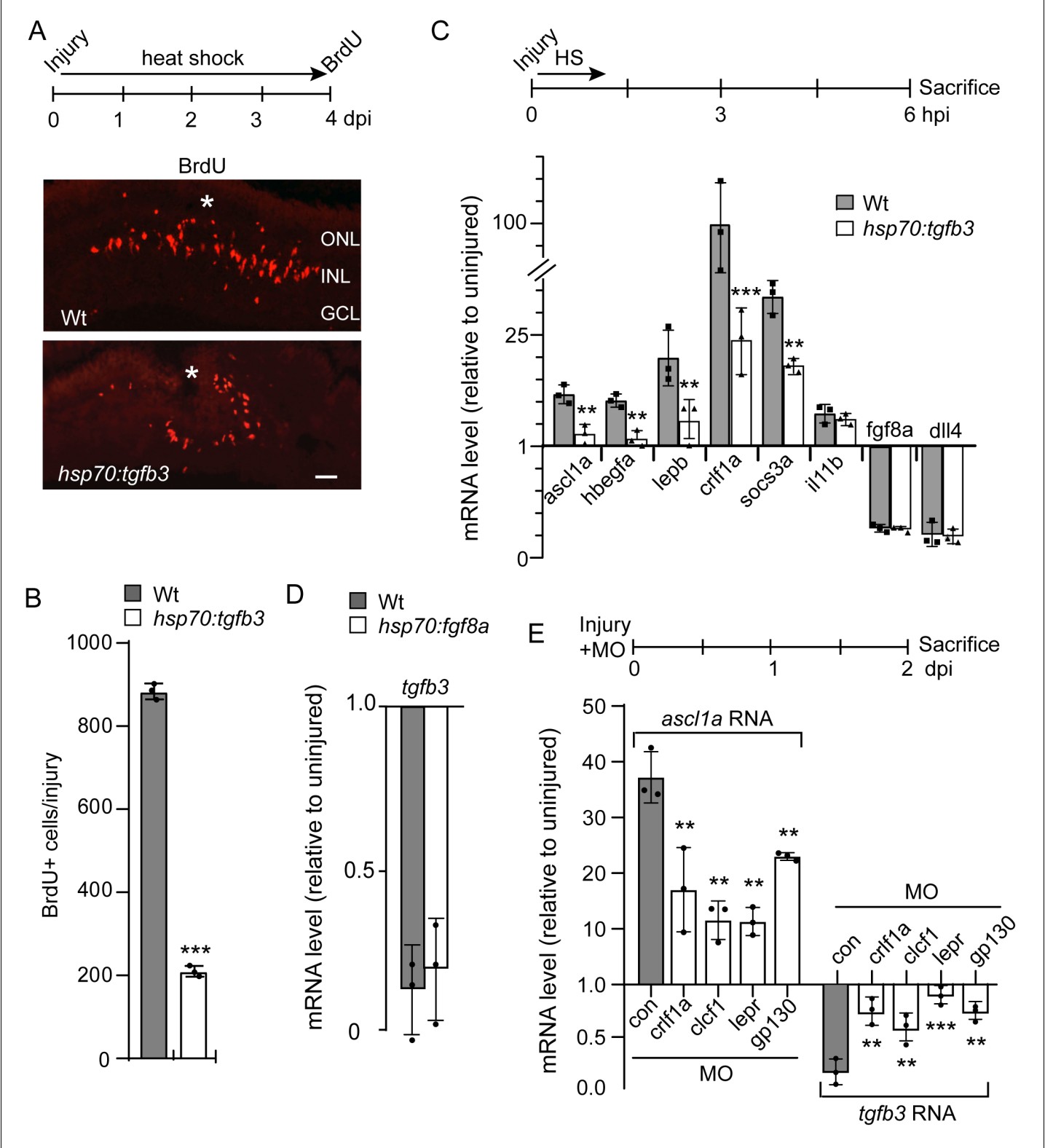

**Figure 4.** Tgfb3 suppresses MG proliferation and reprogramming gene expression. (A) Top illustration is experimental time line. Bottom panels are BrdU immunofluorescence in injured and heat shock-treated Wt and *hsp70:tgfb3* transgenic fish. Asterisk marks injury site. Scale bar is 50 microns. (B) Quantification of data in (A). (C) Top illustration is experimental time line. Fish received a needle poke injury and a 1 hr heat shock and then sacrificed at 6 hpi for RNA analysis by qPCR. Bottom graph is qPCR quantification of select reprogramming gene expression levels in the indicated fish lines at 6 hpi. (D) Experimental time line is as in (C) and *tgfb3* levels were assayed in the indicated fish lines at 6 hpi. (E) Top illustration is experimental time line.
*Figure 4 continued on next page*

*Figure 4 continued*

Bottom panel is qPCR analysis of *ascl1a* and *tgfb3* gene expression with and without the indicated MO treatment. MO, is morpholino-modified antisense oligonucleotide used to knockdown expression from the indicated gene. Error bars are SD. *$p<0.05$, **$p<0.01$, ***$p<0.001$.
The online version of this article includes the following figure supplement(s) for figure 4:

**Figure supplement 1.** *tgfb3* gene expression in *hsp70:tgfb3* fish.

components of these programs are genes encoding growth factors, cytokines, and transcription factors that are rapidly induced following a retinal injury (*Gorsuch et al., 2017*; *Nagashima et al., 2020*; *Nelson et al., 2013*; *Nelson et al., 2012*; *Ramachandran et al., 2010a*; *Ramachandran et al., 2011*; *Ramachandran et al., 2012*; *Thummel et al., 2010*; *Wan and Goldman, 2017*; *Wan et al., 2012*; *Wan et al., 2014*; *Zhao et al., 2014*). Because *tgfb3* expression is suppressed within a few hours post injury, we suspected it might regulate these gene expression programs. To investigate this, we injured Wt or *hsp70:tgfb3* fish retinas and immediately treated fish with a single 1 hr heat shock before sacrificing fish at 6 hpi and isolating retinal RNA for gene expression analysis using qPCR. Interestingly, we found that many reprogramming genes that are normally induced after retinal injury, like *ascl1a*, *hbegfa*, *lepb*, *crlf1a*, and *socs3a*, were suppressed by Tgfb3 expression, while genes normally repressed soon after injury, like *fgf8a* and *dll4* were unaffected (*Figure 4C*). Furthermore, forced Fgf8a expression, which we previously showed suppressed injury-dependent MG proliferation in the adult retina (*Wan and Goldman, 2017*), had no effect on *tgfb3* expression (*Figure 4D*). Thus, Tgfb3 and Fgf8a appear to act independent of each other to regulate MG proliferation.

The observation that *ascl1a* gene expression is suppressed in a Tgfb3-dependent manner was intriguing since it is a critical reprogramming gene that along with Lin28a and Notch inhibition is sufficient to stimulate spontaneous MG proliferation in the uninjured retina (*Elsaeidi et al., 2018*). We previously reported that injury-dependent induction of cytokines, like Crlf1a and Lepb, acting via gp130-coupled receptors, also act upstream of *ascl1a* to drive MG proliferation (*Zhao et al., 2014*). Therefore, we wondered if Tgfb3 was part of this early cytokine response system. For this analysis we took advantage of previously characterized MOs to knockdown cytokine signaling in the injured retina and assayed *ascl1a* and *tgfb3* gene expression (*Figure 4E*; *Zhao et al., 2014*). This analysis confirmed that injury-dependent *ascl1a* gene induction is regulated by a cytokine signaling system, and that this signaling system also contributes to injury-dependent *tgfb3* suppression (*Figure 4E*). Thus, Tgfb3 and cytokine-related gene products seem to contribute to a feedback loop regulating each other's expression and also regulating MG proliferation.

Together, these data suggest Tgfb3 drives MG quiescence in the injured retina, at least in part, by inhibiting the expression of pro-regenerative gene expression programs.

## Tgfb1b overexpression stimulates pSmad3 expression without affecting MG proliferation

qPCR and RNAseq analysis of *tgfb* gene expression in the injured retina revealed that although *tgfb1b* and *tgfb2* were induced in the injured retina, their levels remained below the suppressed levels of *tgfb3* (*Figure 2A*). Furthermore, injury-dependent regulation of pSmad3 expression followed that of *tgfb3*, but not *tgfb1a*, *tgfb1b*, or *tgfb2* (*Figures 1B*, *2A* and *3C*). These observations suggest that Tgfb3 expression largely accounts for pSmad3 expression in the uninjured and injured retina. Whether other Tgfb ligands could also inhibit MG proliferation via a pSmad3 signaling mechanism remained unknown, but seemed likely if we boosted their levels to at least that of Tgfb3 in the uninjured retina. To investigate this, we generated *hsp70:tgfb1b* transgenic fish.

*hsp70:tgfb1b* fish treated with a 1 hr heat shock exhibited over a 100-fold induction of *tgfb1b* RNA that returned to basal levels ~ 24 hr later (*Figure 5—figure supplement 1A*). Overexpression of Tgfb1b had no significant effect on endogenous *tgfb2b* expression, but did decrease *tgfb3* expression by ~50% at 6 hr post heat shock (*Figure 5—figure supplement 1B*). Interestingly, when comparing the consequence of Tgfb1b and Tgfb3 overexpression on MG proliferation and pSmad3 expression, only Tgfb3 suppressed injury-dependent MG proliferation (*Figure 5A–B*), but both Tgfb1b and Tgfb3 reestablished pSmad3 expression at the injury site (*Figure 5C*).

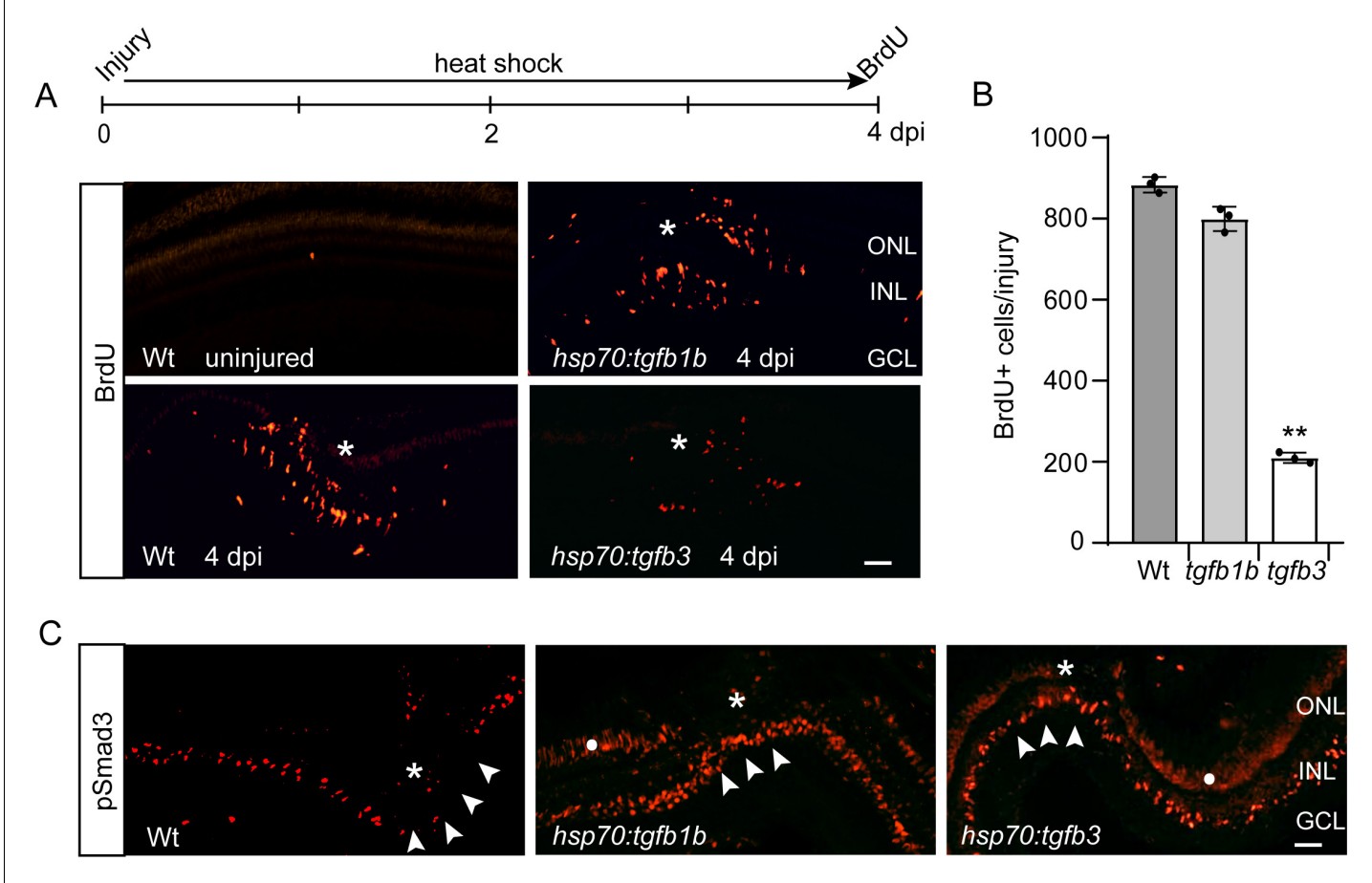

**Figure 5.** Tgfb1b and Tgfb3 stimulate pSmad3 expression, but only Tgfb3 inhibits injury-dependent MG proliferation. (**A**) Top illustration is experimental time line. Bottom panels show BrdU immunofluorescence on retinal sections from uninjured and injured, heat shock-treated Wt, *hsp70: tgfb1b*, and *hsp70:tgfb3* transgenic fish. (**B**) Quantification of data in (**A**). (**C**) pSmad3 immunofluorescence on retinal sections from uninjured and injured, heat shock-treated Wt, *hsp70:tgfb1b*, and *hsp70:tgfb3* transgenic fish. Arrowheads point to the INL at the injury site. Asterisk marks the injury site. White dot in two right-hand panels marks non-specific autofluorescence in the photoreceptor layer. Scale bar is 50 microns. Error bars are SD. **p<0.01.

The online version of this article includes the following figure supplement(s) for figure 5:

**Figure supplement 1.** Heat shock induced *tgfb1b* gene expression in *hsp70:tgfb1b* fish.

Although Tgfb1b overexpression did not regulate MG proliferation, *tgfb1b* RNA is induced in the injured retina (*Figure 2A and C*), and we wondered if this induction had any consequence on regeneration. Therefore, we knocked down Tgfb1b expression with a previously validated splice-blocking *tgfb1b*-targeting MO (*Figure 5—figure supplement 1C*; *Monteiro et al., 2016*). MOs were delivered to retinas at the time of injury and cellular uptake facilitated by electroporation. Four days later, fish received an IP injection of EdU and 3 hr later some fish were sacrificed to assay MG proliferation (*Figure 5—figure supplement 1D*), while others were allowed to survive until 14 dpi to lineage trace EdU+ MG (*Figure 5—figure supplement 1E*). These analyses revealed that Tgfb1b knockdown had no significant effect on injury-dependent MG proliferation or the fate of MG-derived progenitors (*Figure 5—figure supplement 1D–E*).

Together, the above data indicate specificity in the action of Tgfb ligands on MG proliferation and suggest that pSmad3 expression does not drive MG quiescence in the injured retina.

# Alk5 and PP2A inhibition rescue MG proliferation in injured retinas with Tgfb3 overexpression

The above data suggested that Tgfb3-dependent activation of Alk5 may engage a non-canonical Tgfb signaling pathway to regulate MG proliferation. To investigate this, we first determined the effect Alk5 suppression had on MG proliferation in heat shock-treated *hsp70:tgfb3* fish immersed in either DMSO or the Alk5 inhibitor, SB431542 (*Figure 6A*). As expected, Alk5 inhibition rescued MG proliferation in injured retinas from heat shock-treated *hsp70:tgfb3* fish (*Figure 6B–C*).

We next investigated if Tgfb3-dependent activation of Alk5 engaged a non-canonical Tgfb signaling pathway, like ERK, PI3K, p38, and PP2A (*Derynck and Zhang, 2003*; *Petritsch et al., 2000*; *Yu, 2002*; *Zhang, 2017*). Previous studies indicated that ERK and PI3K signaling are necessary for injury-dependent MG proliferation and therefore, are not candidates for conveying the quiescence-promoting effects of Tgfb3 (*Wan et al., 2012*; *Wan et al., 2014*). Therefore, we focused our analysis on PP2A and p38.

PP2A functions as a trimer with a dimeric core consisting of a catalytic (zebrafish genes *ppp2ca*, *ppp2cb*) and structural subunit (zebrafish genes *ppp2r1a*, *ppp2r1b*) and one of 7 regulatory subunits (zebrafish genes *ppp2r2aa*, *ppp2r2ab*, *ppp2r2ba*, *ppp2r2bb*, *ppp2r2ca*, *ppp2r2cb*, *ppp2r2d*) that confers subcellular targeting, substrate specificity, and regulation of holoenzyme phosphatase activity. Importantly, the PP2A Bα subunit (zebrafish Ppp2r2aa and Ppp2r2ab) can associate with and be phosphorylated by Alk5, and this association/phosphorylation is necessary to transduce the anti-proliferative effects of Tgfb receptor signaling (*Griswold-Prenner et al., 1998*; *Petritsch et al., 2000*; *Wlodarchak and Xing, 2016*). Interrogation of RNAseq data sets indicates that genes encoding

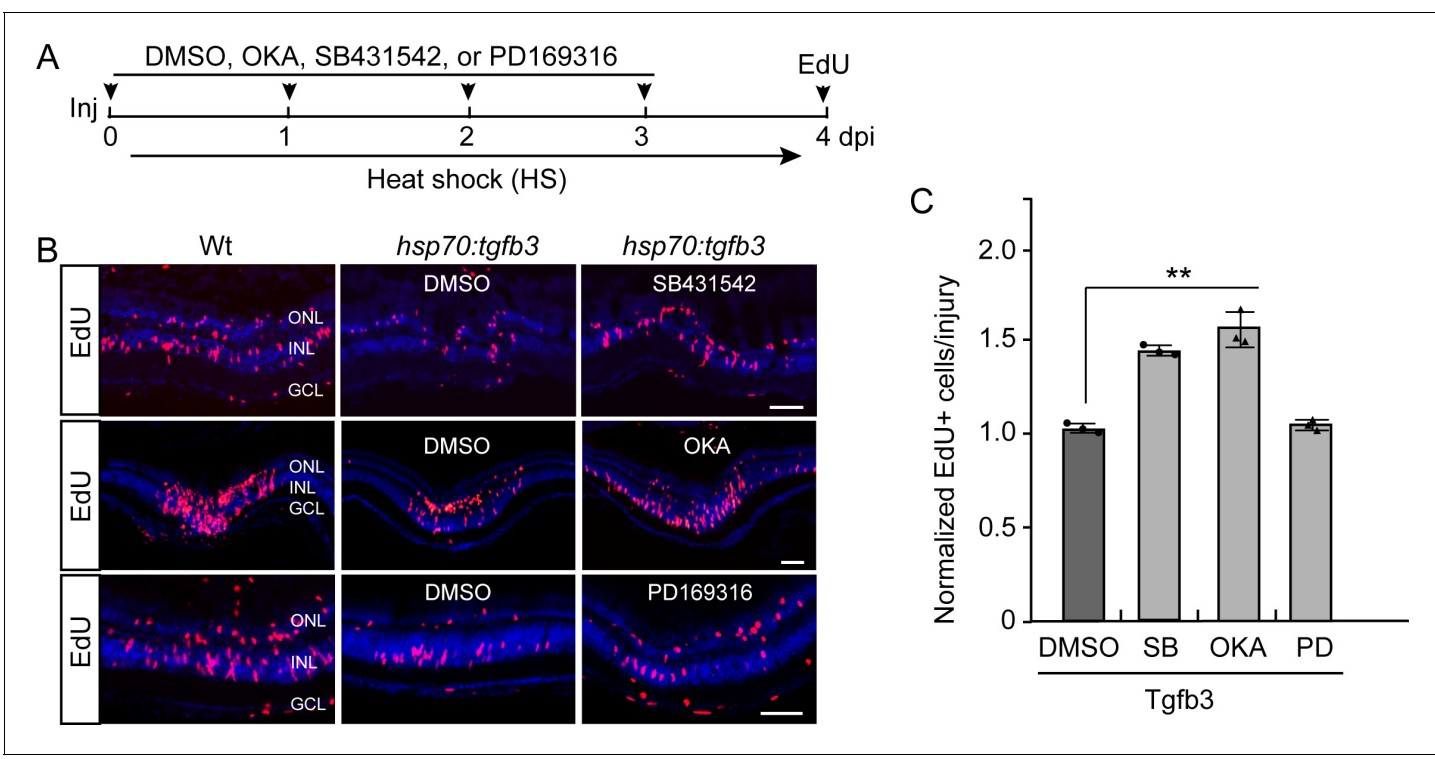

**Figure 6.** Alk5 and PP2A inhibition rescues Tgfb3-mediated inhibition of MG proliferation in the injured retina. (**A**) Experimental time line. (**B**) Edu click chemistry identifies proliferating MG in retinal sections from injured Wt and *hsp70:tgfb3* transgenic fish treated with heat shock, +/- okadaic acid (OKA), PD169316 (PD), or SB431542 (SB) treatment. Scale bar is 50 microns. (**C**) Quantification of the effects of OKA, PD, and SB on Tgfb3-mediated suppression of MG proliferation that is shown in (**B**). Edu values are normalized to MG proliferation in Tgfb3 overexpressing transgenic fish. Error bars are SD. **p<0.01.

The online version of this article includes the following figure supplement(s) for figure 6:

**Figure supplement 1.** Injury-dependent regulation of PP2A subunit and p38 MAPK RNA expression and effect of okadaic acid on cell death in injured retina.

PP2A components and p38 isoforms (*mapk14a*, *mapk14b*) are expressed in quiescent and injury-responsive MG (*Figure 6—figure supplement 1A*).

We next investigated if PP2A or p38 signaling participated in Tgfb3-dependent regulation of MG proliferation in the injured retina. For this analysis, Wt and *hsp70:tgfb3* transgenic fish retinas were injured and fish were either left untreated or received a 1 hr heat shock every 6 hr for 4 days along with daily intravitreal injections of either DMSO, the PP2A inhibitor okadaic acid (10 μM, OKA), or the p38 MAPK inhibitor PD169316 (3 μM, PD) (*Bialojan and Takai, 1988*; *Wilson et al., 1997*). Fish received an IP injection of EdU 3 hr prior to sacrifice to label proliferating cells. Quantification of EdU+ cells revealed that inhibition of PP2A, but not p38 MAPK, rescued MG proliferation in retinas with Tgfb3 overexpression (*Figure 6C*). Neither forced expression of Tgfb3 or intravitreal injection of OKA affected injury-dependent apoptosis (*Figure 6—figure supplement 1B*).

Together, the above data suggests Alk5 and PP2A act downstream of Tgfb3 to regulate MG quiescence.

## Notch inhibition rescues MG proliferation in Tgfb3-expressing injured retinas

Previous studies revealed that Notch signaling inhibition is required for injury-dependent MG proliferation (*Conner et al., 2014*; *Elsaeidi et al., 2018*; *Wan and Goldman, 2017*; *Wan et al., 2012*). Therefore, we wondered if Tgfb3 signaling acted through Notch signaling to inhibit MG proliferation. To investigate this, we bred *hsp70:tgfb3* fish with *tp1:mCherry* Notch reporter fish that harbor 12 RBP-Jk binding sites upstream of a minimal promoter that drives nuclear localized mCherry expression (*Parsons et al., 2009*). Normally, Notch signaling is suppressed in injury-responsive MG spanning the injury site (*Figure 7A*, *tp1:mCherry* panel) (*Elsaeidi et al., 2018*; *Wan and Goldman, 2017*). However, forced Tgfb3 expression in *hsp70:tgfb3;tp1:mCherry* double transgenic fish prevented this injury-dependent suppression in Notch signaling (*Figure 7A–B*) and this correlated with reduced MG proliferation (*Figure 4A–B*). Furthermore, we found that Tgfb3-dependent inhibition of MG proliferation can be rescued by pharmacological suppression of Notch signaling using the γ-secretase inhibitor RO4929097 (*Figure 7C–D*). Consistent with these data, Tgfb3 overexpression in the uninjured retina increased expression of the Notch reporter gene *hey1*, and the ligand encoding gene, *dll4*. However, *dll4* induction did not reach statistical significance (*Figure 7E*) and further studies are needed to determine if Tgfb3 regulates Notch signaling via *dll4* expression. Nonetheless, the above studies suggest Tgfb3 acts, at least in part, via the Notch signaling pathway to regulate MG quiescence.

## Discussion

Here we report that Tgfb3 signaling regulates MG quiescence in the zebrafish retina. Although both Tgfb1b and Tgfb3 can induce pSmad3 expression, only Tgfb3 stimulates MG quiescence, suggesting the involvement of a non-canonical Tgfb signaling pathway. We found that PP2A or Notch inhibition partially rescued MG proliferation in injured retinas overexpressing Tgfb3 and that Tgfb3 acts, at least in part, by stimulating Notch signaling. We also found that Tgfb3-driven MG quiescence is associated with suppression of regeneration-associated genes. Finally, our study reveals that *tgfb3* is highly expressed by pro-regenerative MG of the zebrafish retina, but remains undetectable in non-regenerative MG of the mouse retina.

Tgfb ligands are expressed as latent pre-pro-polypeptides that must be released from latency for their action (*Shi et al., 2011*). Prior to secretion, the immature polypeptide is cleaved between the prodomain and mature peptide domain, but they remain associated. The prodomain is required for proper folding, dimerization, and binding to integrin in the extracellular matrix. In order for Tgfb ligands to signal through their receptors they must be released from the prodomain, which is accomplished via integrin interaction. Among the four different Tgfb ligand encoding genes expressed in the uninjured retina, our data suggests that Tgfb3 predominates. Tgfb3 expression is restricted to quiescent MG and its suppression is required for MG proliferation. Our data suggest that Tgfb3 is constitutively released from integrin in the extracellular matrix and that *tgfb3* gene expression is a major control point in regulating Tgfb signaling in MG.

The mechanism by which *tgfb3* expression is suppressed in the injured retina remains unknown, but diffusible factors emanating from either dying neurons and/or immune cells that accumulate at

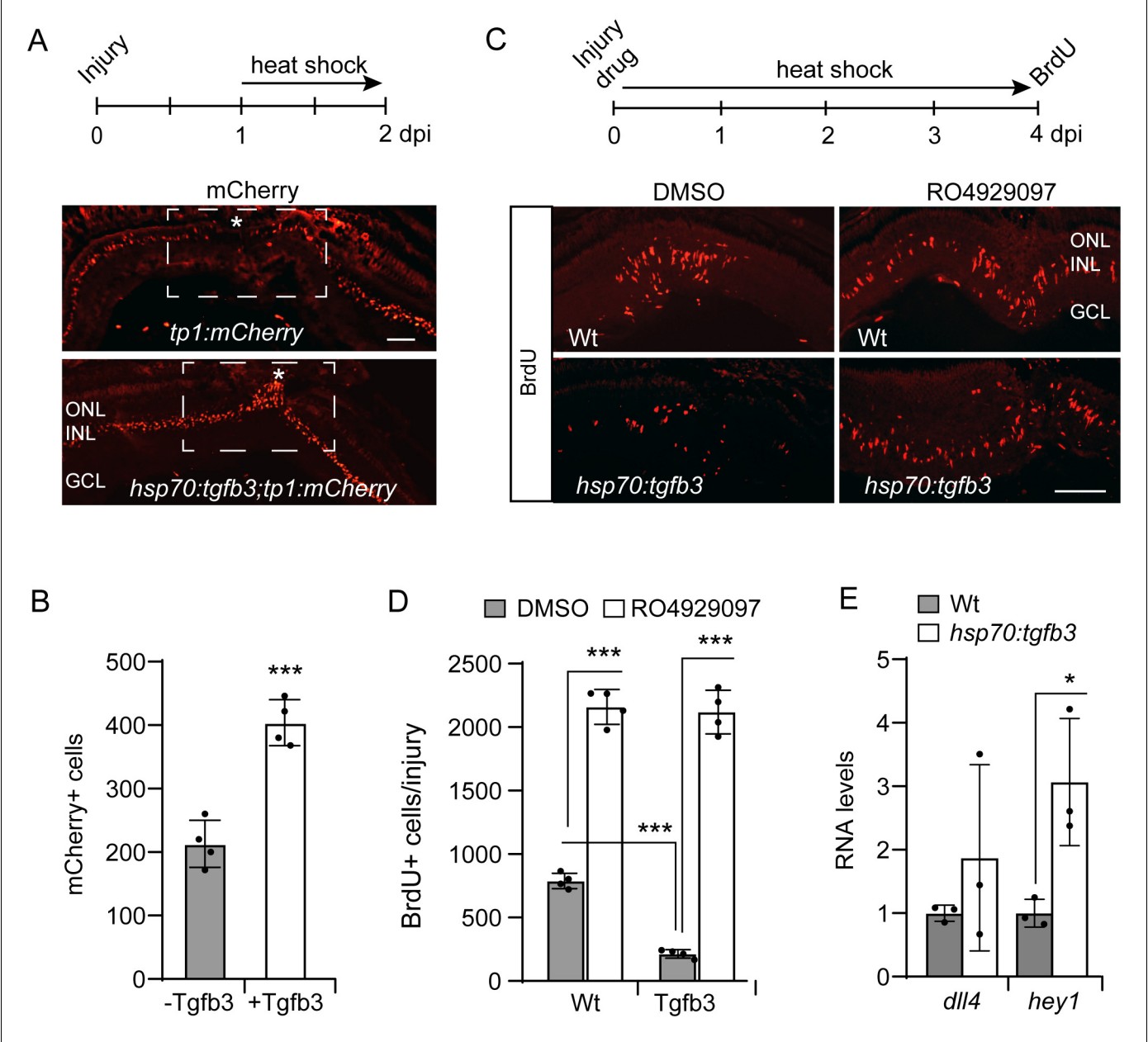

**Figure 7.** Tgfb3 acts upstream of Notch signaling to inhibit MG proliferation. (A) Top illustration is experimental time line. Bottom panels show mCherry immunofluorescence on retinal sections from either injured *tp1:mCherry* or *hsp70:tgfb3;tp1:mCherry* transgenic fish. Asterisk and box region shows the injury site. Note forced Tgfb3 expression stimulate mCherry expression. (B) Quantification of data mCherry+ cells shown in (A). (C) Top is experimental time line. Bottom panels show BrdU immunofluorescence on retinal sections from injured and heat shock-treated Wt and *hsp70:tgfb3* fish treated +/- RO4929097. (D) Quantification of data in (C). (E) qPCR analysis of *dll4* and *hey1* RNA expression using total retinal RNA from Wt and *hsp70: tgfb3* fish that were given a 1 hr heat shock (HS) treatment before sacrifice. Values are normalized to Wt control. Error bars are SD. *p<0.05, ***p<0.001. Scale bar is 50 microns.

the injury site are good candidates. The observation that both the ligand (Tgfb3) and the response (pSmad3) are restricted to MG suggests that Tgfb3 acts in an autocrine or paracrine fashion; however, we cannot rule out the involvement of an intervening neuronal or immune cell. An autocrine/paracrine type of regulation may be a common theme in the injured zebrafish retina since most of the reported secreted factors regulating MG proliferation emanate from MG themselves (*Nelson et al., 2013*; *Wan and Goldman, 2017*; *Wan et al., 2012*; *Zhao et al., 2014*).

In the adult mammalian retina, Tgfb signaling is low and Tgfb expression does not correlate with MG quiescence or proliferation (*Anderson et al., 1995*; *Kugler et al., 2015*; *Lutty et al., 1993*; *Tosi et al., 2018*). However, similar to what we observed in adult fish retina, Tgfb signaling has been associated with reduced MG proliferation in young chicks and postnatal rats (*Close et al., 2005*; *Todd et al., 2017*), and with reduced stem cell proliferation in certain regions of the mouse brain (*Falk et al., 2008*). The high basal level of *tgfb3* in mature MG from zebrafish, but not mice, is intriguing and one cannot help but speculate that it may contribute to their stemness. Interestingly, Notch signaling has been reported to contribute to neural stem cell stemness in the adult zebrafish brain (*Than-Trong et al., 2018*), and in the retina, our data indicates Tgfb3 stimulates Notch signaling. Thus, Tgfb3 expression in the zebrafish retina provides a mechanism for maintaining Notch signaling into adulthood, which is absent in mammals.

Our conclusion that Tgfb3-pSmad3 signaling is active in quiescent MG and suppressed following retinal injury differs from previous reports (*Conedera et al., 2020*; *Lenkowski et al., 2013*; *Sharma et al., 2020*; *Tappeiner et al., 2016*). However, even among these previous studies, inconsistencies emerge, reinforcing the importance of carefully controlled experiments. *Lenkowski et al., 2013* assayed the expression of putative Tgfb responsive genes, like *tgif1* and *tgfbi* to concluded Tgfb signaling was transiently induced prior to injury-dependent MG proliferation and then repressed when MG proliferate. In contrast, *Tappeiner et al., 2016* used pSmad3 immunofluorescence to conclude Tgfb signaling was increased in proliferating MG. Further disparities emerge when examining the effect of the Tgfb signaling inhibitor SB431542 on MG proliferation in the INL where *Tappeiner et al., 2016* reported no effect, and *Sharma et al., 2020* reported inhibition. Remarkably, even in work coming from the same group, inconsistencies emerge. For example, when assaying *tgfb* gene expression, one report suggests increased injury-dependent *tgfb1a* expression, while another indicated reduced *tgfb1a* expression (*Conedera et al., 2020*; *Tappeiner et al., 2016*); and another group reported Tgfb signaling is inhibited in an Oct4-dependent fashion in injury-responsive MG, while a later report indicates enhanced Tgfb signaling in these reprogrammed MG (*Sharma et al., 2019*; *Sharma et al., 2020*). The reason for these disparities is not known, but reinforces the idea that manipulating and assaying Tgfb signaling in the zebrafish retina is not trivial. We also note that many of these studies relied solely on SB431542 to inhibit Tgfb signaling; however, this drug is a more potent inhibitor of CK1 and RIPK2 (*Vogt et al., 2011*), which further clouds the interpretation of results.

In the work reported here, we not only suppressed Tgfb signaling with SB431542 and the more specific Alk5 inhibitor, SB505124 (*Vogt et al., 2011*), but also stimulated Tgfb signaling with conditional expression of Tgfb1b, Tgfb3, and ca-Alk5 using transgenic fish. All these manipulations resulted in the expected change in pSmad3 immunofluorescence in MG, supporting our contention that pSmad3 immunofluorescence reflects Tgfb signaling. Importantly, we found that Tgfb signaling is active in quiescent MG and rapidly suppressed in injury-responsive MG, and that this suppression correlates with *tgfb3* gene expression that was visualized by *in situ* hybridization and quantified by RNAseq and qPCR. Furthermore, unlike previous studies, we determined the relative proportion of RNAs encoding different *tgfb* isoforms, revealing *tgfb3* is expressed at least 80-fold higher than the other isoforms in quiescent MG, and the only isoform to be repressed after retinal injury. These data suggest that Tgfb3 is largely responsible Tgfb signaling in quiescent MG. Remarkably, *Conedera et al., 2020* reported a ~ 2 fold increase in *tgfb3* expression at 1–14 dpi, which is inconsistent with their previous study (*Tappeiner et al., 2016*), and hard to reconcile with our data.

Using transgenic fish, we found that forced expression of Tgfb3, but not Tgfb1b, suppressed injury-dependent MG proliferation. This result contrasts with *Sharma et al., 2020* who reported intravitreal injection of recombinant human Tgfb1 enhanced MG reprogramming and proliferation in the injured. Zebrafish Tgfb1b shares 42% amino acid identity with human TGFb1, and it is not clear if the human ligand will engage zebrafish Tgfb receptors and stimulate pSmad3 expression in the zebrafish retina. Although transgenic approaches used to conditionally express zebrafish ligands are generally preferable to intravitreal injection of human factors, these transgenic approaches do have their limitations since heat shock causes cell stress, and the *hsp70* promoter is induced in all retinal cell types. Our studies suggest that heat shock-induced cell stress had no effect on MG proliferation in wild type and *hsp70:tgfb1b* fish; however, we were unable to achieve conditional, cell-type specific expression of Tgfb3, so it remains possible that some of its actions are indirectly related to its expression in retinal neurons.

One of the more remarkable observations we made during this study is that bothTgfb3 and Tgfb1b can stimulate pSmad3 expression, but only Tgfb3 can drive MG quiescence. This suggested that Tgfb3 must be mediating if effects on MG proliferation via a non-canonical Tgfb signaling pathway. Although the mechanism coupling Tgfb3-dependent activation of Alk5 to MG quiescence remains unknown, our data indicates that it impacts Notch signaling as indicated by increased *tp1: mcherry* transgene and endogenous *hey1* expression.

Experiments designed to rescue MG proliferation in Tgfb3 overexpressing retinas revealed a role for Alk5 and PP2A in regulating MG quiescence. Although it is not surprising that inhibition of Tgfb3's Alk5 receptor would relieve its effects on MG proliferation, the observation that this phenotype is recapitulated by PP2A inhibition was unexpected. Interestingly, PP2A can be recruited to Alk5 to stimulate G1 arrest (*Griswold-Prenner et al., 1998*; *Petritsch et al., 2000*; *Wlodarchak and Xing, 2016*), and PP2A is a major regulator of cell cycle check points and signaling pathways that impinge on the cell cycle, like Wnt, MAPK, PI3K, and mTor (*Wlodarchak and Xing, 2016*). Importantly, these PP2A-regulated signaling pathways have been previously shown to contribute to injury-dependent MG proliferation (*Ramachandran et al., 2011*; *Wan et al., 2012*; *Wan et al., 2014*; *Zelinka et al., 2016*). However, additional studies are needed to determine the mechanism of action of PP2A in regulating MG quiescence and in particular, whether it directly interacts with Alk5.

# Materials and methods

## Key resources table

| Reagent type (species) or resource | Designation | Source or reference | Identifiers | Additional information |
|---|---|---|---|---|
| Strain, strain background (*Danio rerio*) | *1016 tuba1a:GFP* | *Fausett and Goldman, 2006* | | |
| Strain, strain background (*Danio rerio*) | *gfap:GFP* | *Kassen et al., 2007* | | |
| Strain, strain background (*Danio rerio*) | *tp1:mCherry* | *Parsons et al., 2009* | | |
| Strain, strain background (*Danio rerio*) | *zop:nsfb-EGFP* | *Montgomery et al., 2010* | | |
| Strain, strain background (*Danio rerio*) | *hsp70:ca-Alk5* | *Zhou et al., 2011* | | |
| Strain, strain background (*Danio rerio*) | *hsp70:tgfb1b* | This paper; *Figure 5* | | *tgfb1b* expressed under the *hsp70* promoter; generated using Tol2-mediated transgenesis - Goldman lab |
| Strain, strain background (*Danio rerio*) | *hsp70:tgfb3* | This paper; *Figure 4* | | *tgfb3* expressed under the *hsp70* promoter; generated using Tol2-mediated transgenesis – Goldman lab |
| Sequence-based reagent | *Tgfb3*-MO | Gene Tools, LLC | | Lissamine-tagged, *tgfb3*-targeting Morpholino 5'TGCATGGTTAA TATCTGCACACTAT |
| Sequence-based reagent | *Tgfb1b*-MO | Gene Tools, LLC | | Lissamine-tagged, *tgfb1b*-targeting Morpholino 5'AAGGATAGTG CCACTCACTCATTGT |

*Continued on next page*

*Continued*

| Reagent type (species) or resource | Designation | Source or reference | Identifiers | Additional information |
|---|---|---|---|---|
| Sequence-based reagent | T7 universal gRNA primer | Sigma-Aldrich | | T7 universal gRNA primer 5'-AAAAGCACCGACTCGGTG CCACTTTTTCAAGTTGATAAC GGACTAGCCTTATTTTAACTT GCTATTTCTAGCTCTAAAAC-3' |
| Sequence-based reagent | *tgfb3* gRNA one primer | Sigma-Aldrich | | *tgfb3* gRNA one primer: 5'-TAATACGACTCACTAT AGGGCACCTGACTAGGG CCCAGTTTTAGAGCTAGAA |
| Sequence-based reagent | *tgfb3* gRNA two primer | Sigma-Aldrich | | *tgfb3* gRNA two primer: 5'-TAATACGACTCACTAT AGGCCCTCTACAACAGC ACCAGTTTTAGAGCTAGAA |
| Sequence-based reagent | PCR primers | | | See Materials and Methods - Primers and Morpholinos section below |
| Recombinant DNA reagent | *pCS2+ tgfb3-EGFP* | This paper; *Figure 3—figure supplement 1B* | | Vector for generating RNA that has tgfb3 MO target sequence appended to the 5' end of the EGFP mRNA coding sequence - Goldman lab. |
| Recombinant DNA reagent | *pCS2-nCas9n-nanos3'UTR* | Addgene, Plasmid #62542 | Plasmid #62542 | |
| Antibody | anti-pSmad3, rabbit monoclonal | Abcam | Cat. # ab52903 RRID:AB_882596 | 1/200 dilution |
| Antibody | Zpr-1, mouse monoclonal | Zebrafish International Resource Center | Cat. # zpr-1 RRID:AB_10013803 | 1/500 dilution |
| Antibody | Zn-5, mouse monoclonal | Zebrafish International Resource Center | Cat. # zn-5 RRID:AB_10013770 | 1/1000 dilution |
| Antibody | anti-HuC/D, rabbit polyclonal | Abcam | Cat. # ab210554 RRID:AB_210554 | 1/500 dilution |
| Antibody | anti-PKC$_{\beta1}$, mouse monoclonal | Santa Cruz Biotechnology | Cat. # SC-8049 RRID:AB_628143 | 1/200 dilution |
| Antibody | anti-glutamine synthetase (GS), mouse monoclonal | Sigma-Aldrich | Cat. # MAB302 RRID:AB_2110656 | 1/500 dilution |
| Antibody | anti-SOX9, rabbit polyclonal | Millipore Sigma | Cat. # AB5535 RRID:AB_2239761 | 1/500 dilution |
| Antibody | anti-BrdU, rat monoclonal | Thermo Fisher | Cat. # MA 182088 RRID:AB_927214 | 1/500 dilution |
| Antibody | anti-BrdU, mouse monoclonal | Thermo Fisher | Cat. # B35128 RRID:AB_2536432 | Clone MoBu-1 for co-staining with EdU Click-it Chemistry, 1/500 dilution |
| Chemical compound, drug | SB431542 | Fisher Scientific | Cat # 16–141 | Tgfb signaling inhibitor |
| Chemical compound, drug | SB505124 | Fisher Scientific | Cat # 32-631-0 | Tgfb signaling inhibitor |
| Chemical compound, drug | RO4929097 | Cayman Chemical | Cat # 19996 | Notch signaling inhibitor |
| Chemical compound, drug | okadaic acid | Cell Signaling Technology | Cat # 5934 | PP2A inhibitor |
| Chemical compound, drug | PD169316 | Cayman Chemical | Cat # 10006727 | P38 MAPK inhibitor |

*Continued on next page*

*Continued*

| Reagent type (species) or resource | Designation | Source or reference | Identifiers | Additional information |
|---|---|---|---|---|
| Commercial assay or kit | mMESSAGE mMACHINE SP6 Transcription Kit | Invitrogen | Cat # AM1340 | mRNA synthesis |
| Commercial assay or kit | Megascript T7 Transcription Kit | Invitrogen | Cat # AM1334 | mRNA synthesis |
| Commercial assay or kit | In situ cell death, fluorescein | Sigma Aldrich | Cat # 11684795910 | TUNEL assay |

## Animals, injury models, and cell proliferation assays

Animal studies were approved by the University of Michigan's Institutional Animal Care and Use Committee. Zebrafish were kept at 26–28°C with a 10/14 hr light/dark cycle. Adult male and female fish from 6 to 12 months of age were used in these studies. *1016 tuba1a:GFP*, *gfap:GFP*, *tp1: mCherry*, *zop:nsfb-EGFP*, and *hsp70:ca-Alk5* fish were previously described (*Fausett and Goldman, 2006*; *Kassen et al., 2007*; *Montgomery et al., 2010*; *Parsons et al., 2009*; *Zhou et al., 2011*). We generated *hsp70:tgfb1b* and *hsp70:tgfb3* transgenic fish using standard recombinant DNA techniques using Tol2 vector backbone. Expression constructs were injected into single cell zebrafish embryos as previously described (*Fausett and Goldman, 2006*). Fish were anesthetized in tricaine and retinas were injured with a needle poke injury (2–4 injuries/retina for analysis of proliferation and protein expression on retinal sections and 8–10 injuries/retina when harvesting total RNA for qPCR), NMDA, or genetically as previously described (*Fausett and Goldman, 2006*; *Montgomery et al., 2010*; *Powell et al., 2016*). To investigate cell proliferation, fish received an IP injection of BrdU or EdU (10 µl of 10 mg/ml stock) as indicated in the text and detected by immunofluorescence or Click-It chemistry as previously described (*Wan and Goldman, 2017*). Wild-type FVB/N mice were obtained from our breeding colony.

## RNA isolation, PCR, and RNAseq

Total RNA was isolated using Trizol (Invitrogen). cDNA synthesis and PCR reactions were performed as previously described (*Fausett et al., 2008*; *Ramachandran et al., 2010a*). Real-time qPCR reactions were carried out in triplicate with ABsolute SYBR Green Fluorescein Master Mix (Thermo Scientific) on an iCycler real-time PCR detection system (BioRad). The $\Delta\Delta$Ct method was used to determine relative expression of mRNAs in control and injured retinas and normalized to either *gapdh* or *gapdh$_s$* mRNA levels. Individual comparisons were done using unpaired 2-tailed Student t-test. ANOVA with Fisher's PLSD post hoc analysis was used for multiple parameter comparison. Error bars are standard deviation (SD).

For RNAseq, retinas from *1016 tuba1a:GFP* and *gfap:GFP* fish were dissociated and GFP+ MG were purified using FACS using the University of Michigan's Cell Sorting Core as previously described (*Powell et al., 2013*; *Ramachandran et al., 2010a*). RNA from GFP+ cells was used to generate libraries and DNA was sequenced on an Illumina HiSeq2000 instrument. Sequencing reads were analyzed by the University of Michigan's Bioinformatics Core. The number of reads for each expressed gene was determined and differentially expressed genes were restricted to those exhibiting at least a 2-fold difference in expression with threshold abundance greater than 5 Fragments Per Kilobase of transcript per Million mapped reads to eliminate very low abundant transcripts whose estimates of fold-change are unreliable. GEO accession for RNAseq data is GSE145330.

## Generation of *tgfb3* mutant fish

Gene editing was performed as previously described (*Hwang et al., 2013*; *Vejnar et al., 2016*). Briefly, CRISPRscan (https://www.crisprscan.org/) was used to identify gRNA target sequences in exon 1 of the *tgfb3* gene. gRNAs were transcribed using PCR products as a templates and the MEGAscript T7 transcription kit (Thermo Fisher Scientific #AM1334). *Cas9-nanos* mRNA was transcribed using NotI-digested *Cas9-nanos pCS2* expression vector and the mMESSAGE mMACHINE SP6 kit (Invitrogen, #AM1340). *Cas9-nanos* mRNA and two gRNAs targeting *tgfb3* exon1 were co-injected into one cell stage zebrafish embryos. Primers for making *tgfb3* mutant fish: T7 universal

gRNA primer: 5'-AAAAGCACCGACTCGGTGCCACTTTTTCAAGTTGATAACGGACTAGCCTTA
TTTTAACTTGCTATTTCTAGCTCTAAAAC-3'; gRNA one primer: 5'-TAATACGACTCACTATAGGG-
CACCTGACTAGGGCCCAGTTTTAGAGCTAGAA; gRNA two primer: 5'-TAATACGACTCACTA
TAGGCCCTCTACAACAGCACCAGTTTTAGAGCTAGAA.

## Primers and morpholinos (MO) used in this study

The following PCR primers are 5' to 3'. *tgfb3*: forward GATTGGAGGGACGGATGA, reverse GTGA-
CAGGGGCAGTGAAC; *tgfb2*: forward CAGCATGAGAGCCACAGAC, reverse CTCCACAGATACG-
GACAGG; *tgfb1a*: forward GTACAAACACCACAACCCTGG, reverse GGCTTACTTATCAA
TCCCGAC; *tgfb1b*: forward ACTGGCTCTTGCTCCTAT, reverse AACTGTTCCACCTTATGC; *ascl1a*:
forward TTGAGCGTTCGTAAA, reverse GCTGAAGGACTGGATT; *fgf8a*: forward CAGTGTGGA
TACAAACGCAGG, reverse TAGCAAAACGCAAAGAGGTGA; *lepb*: forward CATTGCTCGAAC-
CACCATCAGC, reverse TCTTTATGCACCGGGGTCTCG; *crlf1a*: forward GGGATTCTGGGATCTAG-
GAAAGC, reverse TCCTTGAAGAACCTGGTTGCG; *socs3a*: forward CACTAACTTCTC
TAAAGCAGGG, reverse GGTCTTGAAGTGGTAAAACG; *il-11b*: forward GCTAACAGTGTCGCC
TGACTCC, reverse CTGTAGTTCAGTGAGGGCAGGG; *dll4*: forward GGAAATTTGACGTGCTCCAT,
reverse GAGAAAGGTGAGCCAAGCTG; *hbegf*: forward CGATGGATGGCGAGGATGTAGA, reverse
GCATTAGGGCAGGACGAAGTTG; *hey1*: forward GTTTGCATTTTCACGCCCCT, reverse CGCCCTC
TAGTGCTCACATT; *tgfb3* gRNA mutation forward GGCAAAGGACTGCTGTTTGT, reverse GAGA
TCCCTGGATCATGTTGA; *tgfb1b* MO mis-splicing forward GCACACCATAGAAGATCCAACA,
reverse AGGCATCTGCAACCAGTCTT.

Lissamine-tagged, *tgfb3*-targeting MO: 5'TGCATGGTTAATATCTGCACACTAT; Lissamine-
tagged, *tgfb1b*-targeting MO: 5'AAGGATAGTGCCACTCACTCATTGT; Gene Tools standard control
MO: 5' CCTCTTACCTCAGTTACAATTTATA.

## Morpholino (MO) functional assays

For testing *tgfb3*-targeting MO, we generated *pCS2+tgfb3 EGFP* construct that contained *tgfb3*
cDNA MO target site upstream and in-frame with the EGFP initiator AUG. Primers for generating
*tgfb3* MO targeting sequence are: forward primer (BamHl): 5'-GCAGGATCCGGAGCCGCTTCA
TTCATCTG-3' and reverse primer (Nco1): 5'-TCACCATGGTGGACAGAGACAAGCTCATG-3'. The
*pCS2+tgfb3 EGFP* plasmid was linearized with Not1 restriction enzyme and capped sense *tgfb3-
EGFP* RNA was synthesized using SP6 RNA polymerase using Invitrogen's mMESSAGE mMACHINE
SP6 Transcription Kit (Invitrogen, #AM1340) according to manufactures directions. Following purifi-
cation, the capped *tgfb3-EGFP* RNA was dissolved in nuclease free water containing 0.2% phenol
red and injected with experimental or control MO into single cell zebrafish embryos. Each embryo
received approximately 50 pg of RNA and 250 pg of control or experimental MO. For testing splice
blocking *tgfb1b*-targeting MO, we injected either control MO (2.4 ng) or *tgfb1b*-targeting MO (0.3
ng and 2.4 ng) into single cell zebrafish embryos. RNA was extracted from embryos at 24–48 hr post
injection and assayed for *tgfb1b* mRNA by PCR.

## Heat shock and pharmacological inhibitors

For heat shock, fish were immersed in a water bath at 37°C for 1 hr before returning to system water
at 28°C. For extended periods of heat shock, this was repeated every 6 hr. For inhibition of Tgfb sig-
naling we used two different Tgfb receptor 1 (Alk 5) inhibitors, SB431542 and SB505124 (Fisher Sci-
entific) and to inhibit Notch signaling we used RO4929097 (Cayman). PP2A was inhibited with
okadaic acid (Cell Signaling Technology) and p38 MAPK was inhibited with PD169316 (Cayman
Chemical). Pharmacological reagents were prepared in DMSO as a 10 mM stock and diluted 1/200
in fish water for immersion or PBS for intravitreal injections. Control fish were treated with vehicle.

## Immunofluorescence and *in situ* hybridization

Zebrafish samples were prepared for immunofluorescence as previously described (*Fausett and
Goldman, 2006*; *Ramachandran et al., 2010a*; *Ramachandran et al., 2010a*). Primary antibodies
used in this study: anti-pSmad3, Abcam Cat. # ab52903 (1/200); Zpr-1 and Zn-5, Zebrafish Interna-
tional Resource Center (1/500 and 1/1000, respectively); anti-HuC/D, Abcam, Cat. #ab210554 (1/
500); anti-PKC$_{\beta1}$, Santa Cruz Biotechnology, Cat. #SC-8049 (1/200); anti-glutamine synthetase (GS),

Sigma-Aldrich, Cat. #MAB302 (1/500); anti-SOX9, EMD Millipore, Cat. #AB5535 (1/500); anti-BrdU, Thermo Fisher, Cat. # MA 1–82088 (1/500) and Cat. # B35128 (1/500, clone MoBu-1 for co-staining with EdU Click-It chemistry). Secondary antibodies: Alexa Flour 555 Donkey anti Mouse-IgG (H+L), Thermo Fisher Cat. # A31570 (1:500); Alexa flour 555 Donkey anti Rabbit IgG (H+L), Thermo Fisher, Cat # A31572 (1:500); Alexa flour 555 Donkey anti Sheep IgG (H+L) Thermo Fisher Cat #A21436. Cy3, Jakson Immuno research labs, Cat #712-166-150 (1:500); Alexa Flour 488 donkey anti mouse Thermo Fisher Cat. # A21202 (1:500); Alexa Flour 488 goat anti rabbit Thermo Fisher Cat. # A11008 (1:500); Cy5 goat anti mouse, Thermo Fisher Cat. # A10524 (1:500); and Alexa Flour 647 goat anti rabbit Thermo Fisher Cat # A21244 (1:500). *In situ* hybridization was performed as described previously (*Barthel and Raymond, 2000*).

## Microscopy, TUNEL, cell quantification, and statistical analysis

BrdU and EdU labelling were used to identify and quantify proliferating cells in retinal sections as previously described (*Fausett and Goldman, 2006*; *Ramachandran et al., 2010a*; *Wan and Goldman, 2017*; *Wan et al., 2012*; *Wan et al., 2014*). TUNEL assays were performed on retinal sections using the *in situ* cell death, fluorescein kit (Sigma Aldrich, Cat # 11684795910). Images were captured by a Zeiss Axiophot fluorescence microscope or a Leica DM2500 microscope. Autofluorescence is defined as variable, background fluorescence that can be detected in multiple channels. All experiments were done in triplicate (three trials) with three animals per trial unless otherwise indicated. We routinely quantify the total number of proliferating cells around the injury site in all nuclear layers and also the number of proliferating cells restricted to the INL in order to be sure proliferative changes represent MG proliferation and are not solely due to rod progenitor proliferation. Unless specifically stated, quantification shown in graphs is proliferation in all nuclear layers at the injury site. Quantification of pSmad3 immunofluorescence was restricted to the area surrounding the injury site and represents the field of view. Error bars are standard deviation (SD). ANOVA with Fisher's PLSD *post hoc* analysis was used for multiple parameter comparison; two-tailed Student's *t* test was used for single parameter comparison.

## Acknowledgements

This work was supported by grants from the NIH (NEI RO1 EY018132 and NEI RO1 EY027310) and the Gilbert Family Foundation Vision Restoration Initiative. We thank Caroline Burns, Harvard, for *hsp70:ca-alk5* transgenic fish; David Hyde, Notre Dame, for *zop:nsfb-EGFP* and *gfap:GFP* transgenic fish, Michael Parsons, University California, Irvine, for *tp1:mCherry* transgenic fish, Doyun (George) Kim for help in genotyping fish; Curtis Powell for RNAseq; Flora Rajaei for preliminary *in situ* hybridization assays, Jonathan Jui for help with mouse retinal dissection, Aresh Sahu for suggesting Tgfb3 may regulate notch signaling component expression, Muchu Zhou and Zachary Rekowski for maintaining our zebrafish colony, and all members of the Goldman lab for their comments on this work. We also acknowledge and thank the University of Michigan's Cell Sorting Core, Advanced Genomics Core, and Bioinformatics Core.

## Additional information

### Funding

| Funder | Grant reference number | Author |
|---|---|---|
| Gilbert Family Foundation | Vision Restoration Initiative; AWD011459 | Daniel Goldman |
| National Institutes of Health | NEI RO1 EY018132 | Daniel Goldman |
| National Institutes of Health | NEI RO1 EY027310 | Daniel Goldman |

The funders had no role in study design, data collection and interpretation, or the decision to submit the work for publication.

## Author contributions
Mi-Sun Lee, Jin Wan, Formal analysis, Validation, Investigation, Methodology, Writing - review and editing; Daniel Goldman, Conceptualization, Formal analysis, Supervision, Funding acquisition, Methodology, Writing - original draft, Project administration, Writing - review and editing

## Author ORCIDs
Daniel Goldman (iD) https://orcid.org/0000-0002-0013-1188

## Ethics
Animal experimentation: This study was performed in strict accordance with the recommendations in the Guide for the Care and Use of Laboratory Animals of the National Institutes of Health. Animal studies were approved by the University of Michigan's Institutional Animal Care and Use Committee.

## Decision letter and Author response
Decision letter https://doi.org/10.7554/eLife.55137.sa1
Author response https://doi.org/10.7554/eLife.55137.sa2

# Additional files

## Supplementary files
• Transparent reporting form

## Data availability
GEO accession for RNAseq data is GSE145330.

The following dataset was generated:

| Author(s) | Year | Dataset title | Dataset URL | Database and Identifier |
|---|---|---|---|---|
| Goldman D | 2020 | Injury and apobec2-dependent regulation of zebrafish Muller glial cell gene expression | https://www.ncbi.nlm.nih.gov/geo/query/acc.cgi?acc=GSE145330 | NCBI Gene Expression Omnibus, GSE145330 |

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
