## [Decision Letter]

**Acceptance summary:**

This study nicelydemonstrates that Tgfb3 controls Müller glia quiescence in zebrafish via non-canonical Tgfb signaling. Data also suggest that Notch signaling, and possibly PP2A, act downstream of Tgfb3 which provides additional mechanistic insight into the regenerative process. There is substantial interest in how/why zebrafish possess the capacity to regenerate retinal neurons from Müller glia after injury in comparison to mammals, which cannot. Interestingly, analyses of mouse retinae suggest that Tgfb3 expression is not detectable in mouse Müller glia, which might contribute to their inability to stimulate a regenerative response after retinal injury.

**Decision letter after peer review:**

Thank you for submitting your article "Tgfb3 inhibits retina regeneration via a PP2A-Notch signaling pathway" for consideration by *eLife*. Your article has been reviewed by three peer reviewers, including Jeffrey Gross as the Reviewing Editor and Reviewer #1, and the evaluation has been overseen by Kathryn Cheah as the Senior Editor.

The reviewers have discussed the reviews with one another and the Reviewing Editor has drafted this decision to help you prepare a revised submission.

Summary:

The paper by Goldman and colleague investigates how Tgfb-signaling influences the formation of Müller glia-derived progenitor cells in the fish retina. The authors utilize a variety of experimental manipulations that include state of the art genetic tools, including a constitutively active Alk5, heat-shock-induced Tgfb3, RBP-j/Notch reporter line, and generate a CRISPR-Cas9 deleted *tgfb3^-/-^* fish. Collectively, their data support a model in which resting Müller glia express elevated levels of Tgfb3 and that forced expression of Tgfb3 suppresses the proliferation of progenitors. Furthermore, their data suggest that the Tgfb3-signaling that suppresses proliferation is mediated, in part, through PPP2CA, and possibly also by promoting Notch-signaling.Overall, this is a generally well done study and one that will be of broad interest to the field.

Essential revisions:

1) The link between Tgfb3/pSmad3, PP2A signaling, and the Notch pathway is potentially the most novel aspect of the paper but is not fully developed or characterized. The authors state that Tgfb3-PP2A signaling "impinges" on Notch signaling, but the lack of mechanistic insight into the relationship between these pathways and retina regeneration is disappointing and needs to be experimentally clarified.

2) The findings regarding *ppp2ca* are interesting. However, it seems likely (and is suggested by existing scRNA-seq databases) that *ppp2ca* and all associated catalytic, structural and regulatory subunits are widely expressed in retinal neurons and glia. Thus, intravitreal delivery of okadaic acid is likely to broadly inhibit *ppp2c* in many different retinal cells (and perhaps also have off-target effects). Without targeted inhibition to MG or better descriptions of patterns of expression, the authors should use caution when asserting that *ppp2ca*-mediates non-canonical Tgfb3-signaling to suppress the formation of progenitor cells in damaged fish retinas. Interpretations and conclusions regarding *ppp2ca* should be softened, unless addition targeted experiments are performed that better support this pathway.

3) The findings of current paper have not been adequately discussed in the context of another recent paper from the Ramachandran lab. (Sharma et al., 2020) or Raymond lab (Lenkowski et al., 2013). Many of the findings from the current study are entirely contradictory, despite investigation of similar pathways and use of similar reagents. Particular with respect to patterns of expression of Tgfb isoforms and increases in proliferation of progenitors with the addition of Tgfb1 (Sharma et al., 2020). Additional discussion and explicit comparison of results here is needed.

4) The authors report effective suppression of pSmad3 in resting MG by systemic exposure to Alk5 inhibitors. SB431542 was also used in the study from Sharma and colleagues. It is not clear why the current study did not apply intravitreal SB431542 (similar to okadiac acid or p38 inhibitor) to test whether proliferating of progenitors is influenced by inhibition of Alk5 and, presumptively, failure to up-regulate pSmad3.

5) Normally Tgfbs are secreted as latent factors and require activation by gelatinases. Is it assumed that the *hsp*-driven Tgfb3 is produced as a latent form and then instantly activated by extracellular gelatinases?

6) Only once (for treatment with okadaic acid) are dying cells assessed. Since levels of damage are linked to numbers of proliferating progenitors, it seems prudent to assess levels of cell death with different manipulations to determine whether treatments impact neuronal damage, which in turn might impact numbers of proliferating cells. For example, is there more or less cell death in damaged retinas with loss of tgfb3 or ca-Alk5?

7) For Figure 1D, how were the BrdU cells quantified? It looks like the number of BrdU cells in the INL is similar or even increased, but the number of BrdU cells in the ONL are decreased. Are the cells in the ONL derived from MG? Please clarify.

8) It is surprising that gene expression changes are detectable among Tgfb ligands when using whole retina RNA and the needle poke model. Why was whole retina RNA used in Figure 2? Related, why are Tgfb levels sometimes normalized to Tgfb1 and sometimes to GAPDH?

9) Figure 1—figure supplement 1 – the labeling for pSmad3 in DMSO-treated retinas is strong. However, in inhibitor-treated retinas not only is the nuclear pSmad3 missing from the nuclei of Müller glia, but all of the background labeling is gone, as is autofluorescence in photoreceptor outer segments. This suggests the images were taken at different exposures or post-hoc processing was different. More generally, by what criterion is the fluorescence detected in some of the images referred to as autofluorescence?

10) From a scholarly perspective, the authors are undoubtedly aware that there are many labs doing retina regeneration, yet far too many of the citations only cite work from the Goldman lab. As one simple example, numerous papers have demonstrated a role for Ascl1 (MASH-1) in retina regeneration. A full scholarly account should cite the full breadth of papers across both the fish and mouse models, especially since those findings work together to validate a key role for Ascl1.

11) A few of the analyses lack quantification and statistical analyses (e.g. Figure 1B, C; Figure 7A).

12) All bar graphs should be remade to show individual data points.

---

## [Author Response]

Essential revisions:1) The link between Tgfb3/pSmad3, PP2A signaling, and the Notch pathway is potentially the most novel aspect of the paper but is not fully developed or characterized. The authors state that Tgfb3-PP2A signaling "impinges" on Notch signaling, but the lack of mechanistic insight into the relationship between these pathways and retina regeneration is disappointing and needs to be experimentally clarified.

We have provided additional data (Figure 7E) showing Tgfb3 stimulated hey1 expression which is a Notch-reporter gene in the zebrafish retina (Cell Reports 19: 849-62, 2017). Our data also suggest Tgfb3 may be regulating Notch signaling through its actions on *dll4*. These data, along with the stimulatory effect Tgfb3 has on Notch reporter fish that harbor multiple RBP-Jk binding sites (*tp1:mCherry*) strongly supports the idea that Tgfb3 acts, at least partly, through Notch signaling to regulate MG quiescence. The Results and Discussion section were revised to include our new data and clearly state that additional studies are needed to reveal PP2A mechanism of action (see end of Results and Discussion sections). We also edited the title of the paper for clarity.

We feel that requesting additional experiments to reveal mechanistic details of how Tgfb3 and PP2A mediate their effects on Müller glia proliferation would unreasonably delay publication beyond a year. For example, mechanistic insight into PP2A’s action will require the generation of new transgenic fish that allow for pulldown of Alk5 and PP2A so we can investigate if there is a direct interaction (there are no antibodies that work in fish for this analysis). Production of new transgenic lines and their analysis would take over a year to complete. Because PP2A data is a minor component of the paper (only Figure 6) and because it would take a large amount of time to further investigate its mechanism of action, we prefer to relegate additional mechanistic analysis to a future study.

2) The findings regarding ppp2ca are interesting. However, it seems likely (and is suggested by existing scRNA-seq databases) that ppp2ca and all associated catalytic, structural and regulatory subunits are widely expressed in retinal neurons and glia. Thus, intravitreal delivery of okadaic acid is likely to broadly inhibit ppp2c in many different retinal cells (and perhaps also have off-target effects). Without targeted inhibition to MG or better descriptions of patterns of expression, the authors should use caution when asserting that ppp2ca-mediates non-canonical Tgfb3-signaling to suppress the formation of progenitor cells in damaged fish retinas. Interpretations and conclusions regarding ppp2ca should be softened, unless addition targeted experiments are performed that better support this pathway.

We have softened our statements throughout the paper, including title and Abstract.

3) The findings of current paper have not been adequately discussed in the context of another recent paper from the Ramachandran lab. (Sharma et al., 2020) or Raymond lab (Lenkowski et al., 2013). Many of the findings from the current study are entirely contradictory, despite investigation of similar pathways and use of similar reagents. Particular with respect to patterns of expression of Tgfb isoforms and increases in proliferation of progenitors with the addition of Tgfb1 (Sharma et al., 2020). Additional discussion and explicit comparison of results here is needed.

We address this in the Discussion and also added a little more context on this subject to the Introduction. It is important to realize that even among the published papers and between multiple papers coming from the same group – inconsistencies emerge; we have included this information in the Discussion.

4) The authors report effective suppression of pSmad3 in resting MG by systemic exposure to Alk5 inhibitors. SB431542 was also used in the study from Sharma and colleagues. It is not clear why the current study did not apply intravitreal SB431542 (similar to okadiac acid or p38 inhibitor) to test whether proliferating of progenitors is influenced by inhibition of Alk5 and, presumptively, failure to up-regulate pSmad3.

This experiment was added to Figure 6. Consistent with Alk5 kinase activity being necessary for Tgfb3 effects on MG proliferation, we find SB431542 rescues MG proliferation in the Tgfb3 treated retina. Text was added to Results to describe this result.

5) Normally Tgfbs are secreted as latent factors and require activation by gelatinases. Is it assumed that the hsp-driven Tgfb3 is produced as a latent form and then instantly activated by extracellular gelatinases?

We added to the second paragraph of the Discussion the mechanism underlying Tgfb activation. Prodomain cleavage takes place in golgi prior to secretion and the non-covalent complex of latency domain and mature peptide bind integrin in the extracellular matrix which stimulates release of the mature protein so it can interact with its receptor. These proteins don’t require extracellular gelatinases for their activity.

6) Only once (for treatment with okadaic acid) are dying cells assessed. Since levels of damage are linked to numbers of proliferating progenitors, it seems prudent to assess levels of cell death with different manipulations to determine whether treatments impact neuronal damage, which in turn might impact numbers of proliferating cells. For example, is there more or less cell death in damaged retinas with loss of tgfb3 or ca-Alk5?

We include the TUNEL data for ca-Alk5 overexpression and Tgfb3 loss in Figure 1—figure supplement 1B and Figure 4—figure supplement 1C, respectively. We found ca-Alk5 suppressed TUNEL+ cell number in the injured retina, while Tgfb3 loss had no effect. We comment on the ca-Alk5 effect on TUNEL+ cells at the end of the first section of the Results under the heading “pSmad3 signaling is suppressed…” Our original data, presented in Figure 6—figure supplement 1B, showed Tgfb3 overexpression and okadaic acid treatment had no effect on TUNEL+ cells in the injured retina.

7) For Figure 1D, how were the BrdU cells quantified? It looks like the number of BrdU cells in the INL is similar or even increased, but the number of BrdU cells in the ONL are decreased. Are the cells in the ONL derived from MG? Please clarify.

Although MG nuclei migrate towards the ONL and divide at the base of the ONL, MG-derived progenitors divide in the INL. Thus, counting all proliferating cells is appropriate. Because we use a needle poke injury, proliferating cells at the site of injury are often displaced towards the GCL, while proliferating cells flanking the injury site are not displaced. This is what is observed in this picture. The reviewer is only looking at the displaced cells in the top Wt panel not the cells restricted to the INL. To help clarify this we labelled the various layers in the top panel. We added a couple of sentences to the Materials and methods subsection “Microscopy, cell quantification and statistical analysis”, to clarify how we do our counts and what is reported.

8) It is surprising that gene expression changes are detectable among Tgfb ligands when using whole retina RNA and the needle poke model. Why was whole retina RNA used in Figure 2? Related, why are Tgfb levels sometimes normalized to Tgfb1 and sometimes to GAPDH?

Figure 2A and B use RNA from FACS purified GFP+ MG from gfap:GFP and 1016 tuba1a:GFP fish and we added this to the figure legend (it is also in the Materials and methods section). We used total RNA from retinas for the time course experiments since it would be too costly and complex to FACS purify MG for every time point. However, the reason we can detect gene expression changes is that we give the fish 8-10 needle pokes when we analyze total retinal RNA by qPCR, which activates a regenerative response in most MG. This information was added to the Materials and methods subsection – “Animals, injury models, and cell proliferation assays”. All samples are corrected for variations in RNA amount using gapdh as a normalizer; however, to illustrate the magnitude of *tgfb3* RNA abundance relative to the other injury-regulated *tgfb* genes, we normalized to *tgfb1b* which is the least abundant *tgfb* ligand encoding gene in MG.

9) Figure 1—figure supplement 1 – the labeling for pSmad3 in DMSO-treated retinas is strong. However, in inhibitor-treated retinas not only is the nuclear pSmad3 missing from the nuclei of Müller glia, but all of the background labeling is gone, as is autofluorescence in photoreceptor outer segments. This suggests the images were taken at different exposures or post-hoc processing was different. More generally, by what criterion is the fluorescence detected in some of the images referred to as autofluorescence?

We reimaged the sections and provide a new figure. Autofluorescence is defined as variable, background fluorescence that can be detected in multiple channels. This statement was added to the Materials and methods section.

10) From a scholarly perspective, the authors are undoubtedly aware that there are many labs doing retina regeneration, yet far too many of the citations only cite work from the Goldman lab. As one simple example, numerous papers have demonstrated a role for Ascl1 (MASH-1) in retina regeneration. A full scholarly account should cite the full breadth of papers across both the fish and mouse models, especially since those findings work together to validate a key role for Ascl1.

We added references as suggested.

11) A few of the analyses lack quantification and statistical analyses (e.g. Figure 1B, C; Figure 7A).

Quantification has been added to these figures.

12) All bar graphs should be remade to show individual data points.

Graphs were remade as requested with individual data points shown.